# Thermoelectric signature of the chiral anomaly in $Cd_3As_2$

Zhenzhao Jia[1], Caizhen Li[1], Xinqi Li[1], Junren Shi[2,3], Zhimin Liao[1,3], Dapeng Yu[1,3] & Xiaosong Wu[1,3]

Discovery of Weyl semimetals has revived interest in Weyl fermions which has not been observed in high energy experiments. It now becomes possible to study, in solids, their exotic properties. Extensive photoemission spectroscopy and electrical resistivity experiments have been carried out. However, many other properties remain unexplored. Here we show the thermoelectric signature of the chiral anomaly of Weyl fermions in $Cd_3As_2$ under a magnetic field. We observe a strong quadratic suppression of the thermopower when the magnetic field is parallel to the temperature gradient. The quadratic coefficient is nearly twice of that for the electrical conductivity. The thermopower reverses its sign in high fields. We show that all these intriguing observations can be understood in terms of the chiral anomaly of Weyl fermions. Our results reveal the anomalous thermoelectric property of Weyl fermions and provide insight into the chiral anomaly.

[1] State Key Laboratory for Artificial Microstructure and Mesoscopic Physics, Department of Physics, Peking University, Beijing 100871, China. [2] International Center for Quantum Materials, Peking University, Beijing 100871, China. [3] Collaborative Innovation Center of Quantum Matter, Beijing 100871, China. Correspondence and requests for materials should be addressed to Z.L. (email: liaozm@pku.edu.cn) or to X.W. (email: xswu@pku.edu.cn).

Recently experimental progress on theoretically predicted Weyl semimetals has stirred strong interest in Weyl fermions among condensed matter physicists[1–5]. Weyl semimetals are three-dimensional quantum materials whose conduction band and valence band touch at individual points in the Brillouin zone, so-called Weyl nodes. Besides a linear dispersion in the vicinity of nodes, electron states are also chiral, indicated by the Hamiltonian, $H = \chi v \mathbf{p} \cdot \boldsymbol{\sigma}$. Here, $v$, $\mathbf{p}$ and $\sigma_i$ are the velocity, momentum and Pauli matrices, respectively. $\chi = \pm 1$ labels the chirality of electrons, associated with which is one of the most exotic properties of Weyl fermions, the chiral anomaly[6–11]. Intensive efforts have been made in observing the related effects. One of the consequences of the chiral anomaly, a negative longitudinal magnetoresistance (MR), has been under the spotlight and observed in a flood of experiments on various materials[12–21].

Despite a host of other novel effects that have been predicted for Weyl fermions due to the chiral anomaly[22–35], few experiment has been reported. Complementary to the electrical resistivity, the thermoelectric effect provides unique information on the electronic transport and has been used to study two-dimensional massless Dirac fermions in graphene and three-dimensional massive Dirac fermions[36–40]. It is associated with the derivative of the electric conductivity with respect to energy. Therefore, it highlights the energy dependence of the transport.

In this work, we study the thermoelectric effect of a Dirac semimetal $Cd_3As_2$, which becomes a Weyl semimetal when the time reversal symmetry is broken by a magnetic field $B$. Under a field that is parallel to the temperature gradient $\nabla T$, we observe a negative magneto-thermopower, which is quadratic in small fields. The $B^2$ coefficient is nearly twice of that for the field dependence of the electrical conductivity. Intriguingly, the thermopower reverses its sign at high fields. Based on the chiral anomaly and the Mott relation, we derive a simple formula, which explains these observations. Our experiments reveal the anomalous thermoelectric property of Weyl fermions and provide insight into the chiral anomaly.

## Results

**Temperature dependence of resistivity and thermopower.** The device structure is shown in the inset of Fig. 1a. The measurement set-up was borrowed from previous work by Small et al.[41]. The armchair-like metal line on the bottom serves as a micro-heater,

which can generate a temperature gradient along the vertical direction when passing a current. The electrodes on the two ends of the sample are four-probe resistive thermometers. They are also current leads for the electrical resistance measurement and voltage probes for thermopower detection. An a.c. method was employed to measure the temperature difference $\Delta T$ and the thermoelectric voltages. The detailed description of the measurements can be found in Supplementary Fig. 1 and Supplementary Note 1.

The temperature dependence of the resistivity $\rho$ for a 700 nm thick $Cd_3As_2$ platelet is plotted in Fig. 1a. With increasing temperature, the resistivity increases first and then starts to decrease at $\sim 30$ K. This behaviour is commonly seen in semimetals of low carrier concentrations[13,18]. At low temperatures, the carrier concentration is relatively constant and the resistivity exhibits a metallic temperature dependence. As the temperature is raised, more carriers are thermally activated, which leads to reduction of resistivity. The low carrier concentration of our nanowire and micro-plate samples enables us to observe a strong chiral anomaly induced negative MR[14]. The temperature dependence of the thermopower $S_{xx}$, shown in Fig. 1b, confirms the thermal activation process. At low temperatures, $S_{xx} > 0$, indicting that holes are dominant carriers, while above 57 K, $S_{xx}$ becomes negative, indicating that thermally activated electrons become dominant. This can be qualitatively understood by a two-band model. Assuming the electrical conductivity for two bands are $\sigma_1$ and $\sigma_2$, and the thermopower are $S_1$ and $S_2$, the total thermopower would be a weighted sum of $S_1$ and $S_2$, $S = \frac{\sigma_1 S_1 + \sigma_2 S_2}{\sigma_1 + \sigma_2}$. Since the electron mobility is much higher than the hole mobility in $Cd_3As_2$ (refs 42,43), its conductivity will exceed that of holes as the temperature increases. So, its contribution to $S$ will eventually dominate. The switching of the dominant carrier is also evident in our Hall measurement (Supplementary Fig. 2 and Supplementary Note 2).

**Magnetotransport in a perpendicular field.** When a magnetic field perpendicular to the plate is applied, the resistivity is substantially enhanced, seen in Fig. 2. As the temperature increases, MR, defined as $\rho(B)/\rho(0) - 1$, increases, reaching 1,800% at $T = 150$ K. A large positive MR is characteristic in $Cd_3As_2$ (refs 44–46). Shubnikov-de Haas oscillations are not discernible, likely due to the low mobility and low density of the hole band[14,18]. The thermopower $S_{xx}$ exhibits similar enhancement with field at low temperatures, except for a

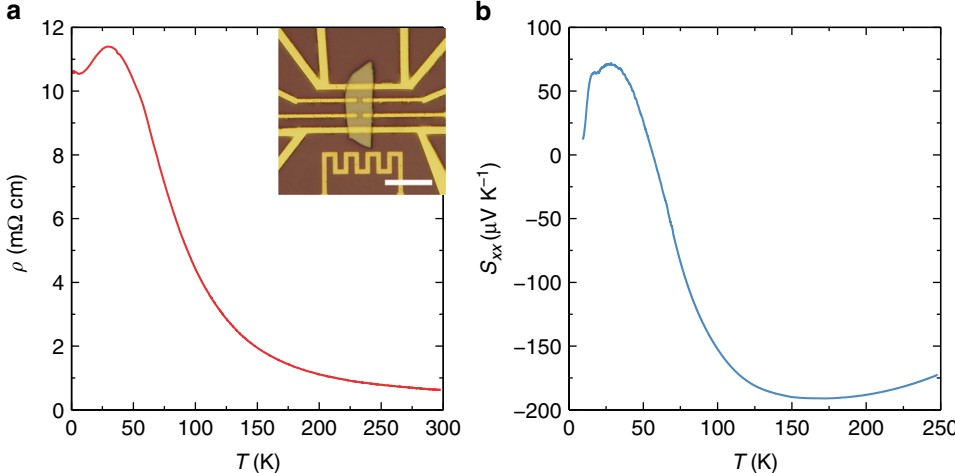

**Figure 1 | Temperature dependence of $\rho$ and $S_{xx}$.** (a) $\rho$ as a function of temperature. Inset, an optical micrograph of a device. The scale bar is 20 μm. (b) Thermopower $S_{xx}$ as a function of temperature.

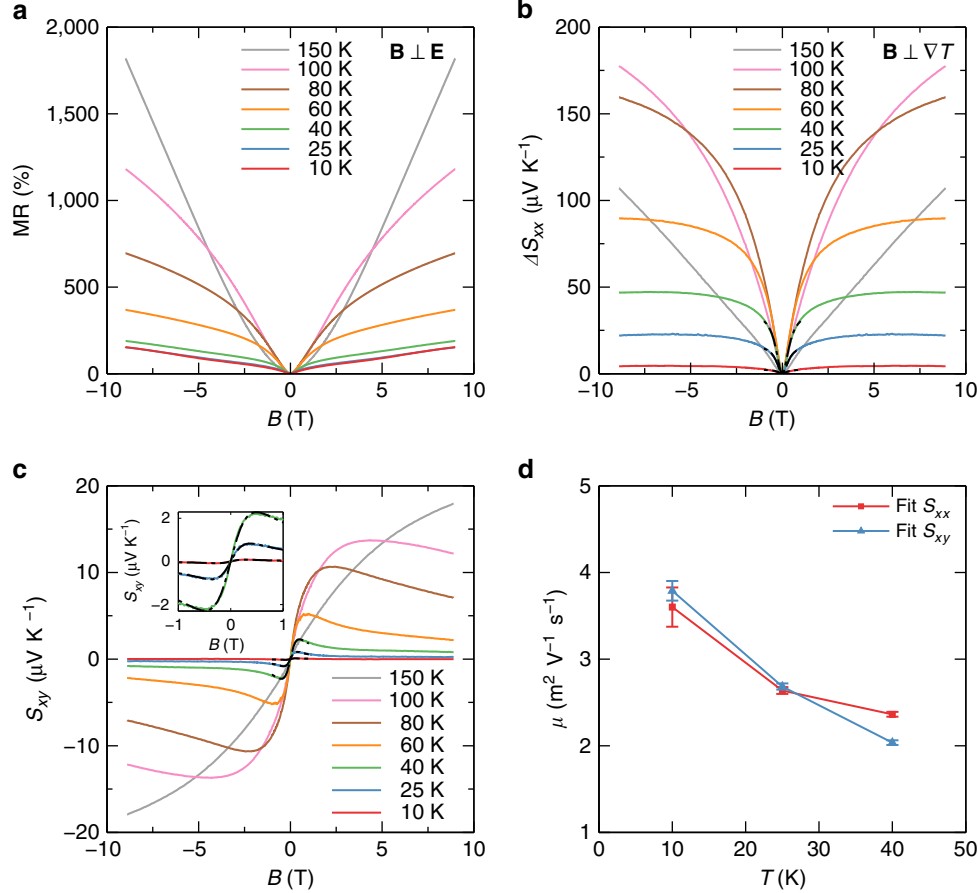

**Figure 2 | Resistivity and thermoelectric effects in a perpendicular field.** (**a**) Large positive MR at various temperatures. (**b**) Change of the thermopower with the magnetic field $\Delta S_{xx} = S_{xx}(B) - S_{xx}(0)$. (**c**) Nernst effect $S_{xy}$ versus the magnetic field. The black dash–dot lines are fits to equations (2) and (3) at $T = 10$, 25 and 40 K. The inset in **c** is a zoom-in plot. (**d**) Fitted mobility $\mu$ versus temperature. Red squares are obtained from $S_{xx}$ and blue triangles are from $S_{xy}$. The error bars represent the 95% confidence intervals of fits.

saturation in high fields, as plotted in Fig. 2b. Similar field dependence of the thermopower has been reported for three-dimensional massive Dirac states in $Pb_{1-x}Sn_xSe$ (ref. 40). It has been shown that both $S_{xx}$ and $S_{xy}$ can be explained by a single-band Boltzmann-Drude transport combined with the Mott relation[40]. The Mott relation relates the thermoelectric conductivity $\boldsymbol{\alpha}$ with the derivative of the electrical conductivity $\boldsymbol{\sigma}$,

$$\alpha_{ij} = \frac{\pi^2 k_B^2 T}{3e} \frac{\partial \sigma_{ij}}{\partial \varepsilon}\bigg|_{\varepsilon = E_F}, \quad i,j = x, y, \quad (1)$$

where $k_B$ is Boltzmann constant, $e$ the elementary charge, $T$ the temperature, $\varepsilon$ the energy and $E_F$ the chemical potential. From $\alpha_{ij}$, we derive the thermopower and Nernst effect[40],

$$S_{xx} = \frac{\pi^2 k_B^2 T}{3e}\left(\frac{\sigma_{xx}^2}{\sigma_{xx}^2 + \sigma_{xy}^2}D + \frac{\sigma_{xy}^2}{\sigma_{xx}^2 + \sigma_{xy}^2}D_H\right), \quad (2)$$

$$S_{xy}(B) = \frac{\pi^2 k_B^2 T}{3e}\frac{\sigma_{xx}\sigma_{xy}}{\sigma_{xx}^2 + \sigma_{xy}^2}(D_H - D), \quad (3)$$

where $D = \partial \ln \sigma_{xx}/\partial \varepsilon$ and $D_H = \partial \ln \sigma_{xy}/\partial \varepsilon$ at $\varepsilon = E_F$. We adopt $\sigma_{xx} = n_p e \mu/(1 + \mu^2 B^2)$ and $\sigma_{xy} = n_p e \mu^2 B/(1 + \mu^2 B^2)$ for a single-band Boltzmann-Drude transport to calculate $D$ and $D_H$, where $\mu = e v_F \tau/\hbar k_F$ is the mobility for massless Dirac fermions. Here, $v_F$ and $k_F$ are the Fermi velocity and the Fermi wave vector, respectively. $\tau$ is the mean free time. $n_p = 3.8 \times 10^{17}\,cm^{-3}$ is estimated from the Hall resistance. Taking $v_F = 0.5 \times 10^6\,m\,s^{-1}$

for the valence band and assuming $\tau$ is independent of energy (Supplementary Note 3), we fit our data to equations (2) and (3). The fits were only performed for curves measured at low temperatures where $S_{xx}$ does not change its sign. At higher temperatures, both electron and hole bands will need to be considered. The complexity of the two-band model for the thermoelectric effect prevents a straightforward analysis and therefore throughout this work, data for thermoelectric effect above $T = 50$ K are left out in detailed analysis. As depicted in Fig. 2b,c, the fits are reasonably good. Furthermore, both fits yield a similar $\mu$, which is plotted in Fig. 2d. It is worth mentioning that taking into consideration a magnetic dependent $\tau$, as suggested by the linear MR of $Cd_3As_2$ in Fig. 2a, only slightly changes the fittings (Supplementary Fig. 3 and Supplementary Note 4).

**Magnetotransport in a parallel field.** We now turn to the transport when the magnetic field is parallel to the electric field (which is also the direction of the temperature gradient in the thermoelectric measurement), $\mathbf{B}\|\mathbf{E}$. It is known that a magnetic field breaks the time reversal symmetry and splits the Dirac node of a three-dimensional Dirac semimetal into two Weyl nodes in the momentum space along the field direction[47,48]. Thus, the Dirac semimetal turns into a Weyl semimetal. This has been experimentally demonstrated in $Cd_3As_2$ (refs 14,18). When the magnetic field is parallel to the electric field, the chiral anomaly gives rise to a negative MR. We have indeed observed a strong

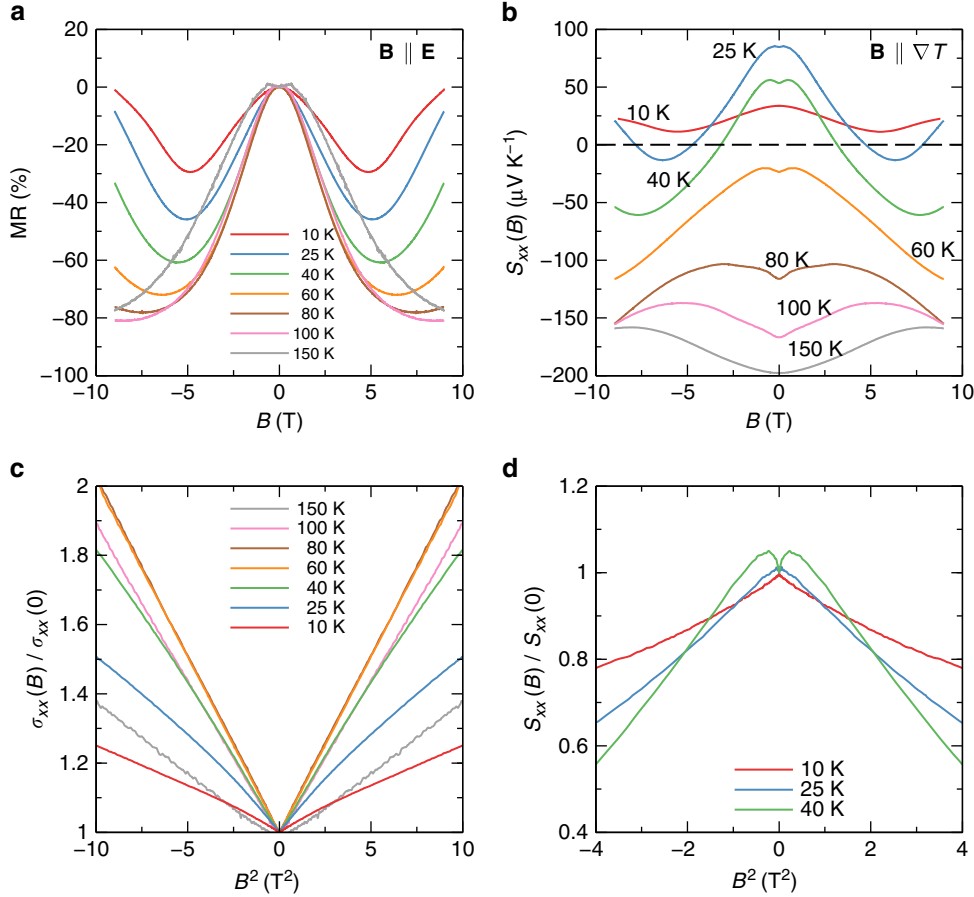

**Figure 3 | Resistivity and thermoelectric effects in a parallel field.** (**a**) Large negative MR at various temperatures. (**b**) Thermopower $S_{xx}$ versus the magnetic field. (**c**) Conductivity versus $B^2$. (**d**) Thermopower versus $B^2$.

negative MR in our samples. Shown in Fig. 3a, the resistivity drops with $B$. When the temperature increases, the negative MR becomes even stronger. It is over 80% at 100 K. At low temperatures, the resistivity increases in high fields. All these behaviours have been discussed in our previous studies[14] and consistent with others[18].

The chiral anomaly leads to a conductivity that depends quadratically on $B$ in the low field limit, described by[9]

$$\Delta\sigma_{xx}^{C} = \frac{e^4 v_F^3 \tau_i B^2}{2\pi^2 \hbar E_F^2}, \tag{4}$$

where $\tau_i$ is the intervalley scattering time and we have taken the valley degeneracy of 2 for $Cd_3As_2$ into consideration. By plotting $\sigma_{xx}(B)$ against $B^2$, curves for all temperatures are linear when $B$ is not large, in a good agreement with the chiral anomaly. The 150 K curve remains linear up to 9 T (not shown). Note that emergence of electron carriers due to thermal excitation will not change the quadratic dependence of $\sigma_{xx}$, as the total conductivity is a simple sum of two types of carriers.

At low temperatures ($T = 10$, 25 and 40 K) where one hole band dominates the transport, the thermopower decreases with the magnetic field. $S_{xx}$ even drops to zero and change its sign. This anomalous field dependence is in sharp contrast to that in a perpendicular field, which we have shown can be explained by a dominant single band. Thus, the competition between two types of carriers is excluded as the origin of the anomalous field dependence. In Fig. 3d, $S_{xx}$ in low fields is plotted against $B^2$. It shows a quadratic field dependence, too. The coefficients of the quadratic dependence for $\sigma_{xx}(C_{MR})$ and $S_{xx}(C_{MS})$ and their ratio

**Table 1 | The coefficients of the $B^2$ dependence for $\sigma_{xx}$ and $S_{xx}$.**

| $T$ (K) | $C_{MR}$ (/T$^2$) | $-C_{MS}$ (/T$^2$) | $-C_{MS}/C_{MR}$ | $B_0$ (T) | $\tau_i/\tau$ |
|---|---|---|---|---|---|
| 10 | 0.0255 | 0.0489 | 1.92 | 6.27 | 14.5 |
| 25 | 0.0527 | 0.0899 | 1.71 | 4.36 | 30.1 |
| 40 | 0.0844 | 0.0139 | 1.65 | 3.44 | 48.1 |

are summarized in Table 1. Interestingly, the ratio $-C_{MS}/C_{MR}$ is close to 2.

**Chiral anomaly induced thermopower.** To understand these intriguing observations for $S_{xx}$, we apply equation (2) to the parallel field configuration. Because all off-diagonal terms disappear here, equation (2) is reduced to the common form of the Mott relation, $S_{xx} = \frac{\pi^2 k_B^2 T}{3e} \frac{\partial \ln \sigma_{xx}}{\partial \varepsilon}\big|_{\varepsilon=E_F}$. Since the total conductivity is given by the sum of the Drude conductivity $\sigma_{xx}^0 = n_p e\mu = \frac{2e^2\tau E_F^2}{3\pi^2 \hbar^3 v_F}$ and $\Delta\sigma_{xx}^C$ due to the chiral anomaly, we have

$$\frac{\sigma_{xx}(B)}{\sigma_{xx}(0)} = 1 + B^2/B_0^2, \tag{5}$$

where $B_0 = \sqrt{3\tau/\tau_i} B_q$. $B_q = 2E_F^2/3e\hbar v_F^2$ is the quantum magnetic field for massless Dirac fermions in $Cd_3As_2$. Plugging

equation (5) into the Mott relation yields

$$\frac{S_{xx}(B)}{S_{xx}(0)} = \frac{1 - B^2/B_0^2}{1 + B^2/B_0^2}. \tag{6}$$

When $B$ is small, equation (6) is further reduced to

$$\frac{S_{xx}(B)}{S_{xx}(0)} \approx 1 - 2B^2/B_0^2. \tag{7}$$

Equation (7) agrees well with the $B^2$ dependence of our result, shown in Fig. 3d. Moreover, it accounts for the observed ratio of near 2 in the $B^2$ coefficients for $\sigma_{xx}$ and $S_{xx}$, which strongly suggests that our analysis captures the essential physics for the chiral anomaly induced thermopower.

By fitting $\sigma_{xx}(B)$ to equation (5), $B_0$ is determined. Then, the ratio of $\tau_i/\tau$ can be calculated, listed in Table 1. It is in the range of 14.5–48.1, consistent with a recent study[18]. Note that a large $\tau_i/\tau$ is also required for observation of the chiral anomaly[9,10]. Using the obtained $B_0$, we fit the low temperature $S_{xx}(B)$ curves to equation (6) without any free parameter, except for the one at $T = 40\,\mathrm{K}$, for which a prefactor is introduced to account for the dip around zero magnetic field. Surprisingly, equation (6) reproduces the experiment very well in a large field range, seen in Fig. 4. Similar results have been obtained in other samples too (Supplementary Figs 4 and 5 and Supplementary Note 5).

We notice a dip in $S_{xx}$ around zero field, see Fig. 3b. It is absent at 10 K, while it emerges at 25 K and grows with temperature. The evolution is consistent with the contribution from thermally activated electrons. It is reasonable to believe that electrons also display the chiral anomaly induced negative magneto-thermopower. Due to the low density and high mobility, their contribution will be prominent at low fields, forming a dip around zero field. As the temperature increases, electrons gradually gain its weight and eventually dominate $S_{xx}$. At 150 K, $S_{xx}$ recovers a $B^2$ dependence, in agreement with equation (7). However, the coefficient ratio $-C_{\mathrm{MS}}/C_{\mathrm{MR}}$ is 0.2, significantly <2. The reason is not clear, but it might be related to thermal smearing of the chiral anomaly or electron phonon scattering (Supplementary Fig. 6 and Supplementary Note 6).

## Discussion

The chiral anomaly conductivity is predicted to be proportional to $B^2$ and inversely proportional to $E_F^2$. The $B^2$ dependence has been confirmed in various Dirac semimetals and Weyl semimetals[13,15,18]. However, it is difficult to study the energy

(or density) dependence, as gating is generally non-uniform for three-dimensional materials because of screening. Thermoelectric effects offers a technique to address this issue, as it is proportional to the derivative of the conductivity with respect to energy. The inverse proportionality of conductivity to energy leads to the negative sign of the second term on the right-hand side of equation (6), hence a suppression of the thermopower with $B$. When this chiral term is larger than the conventional Drude term, $S_{xx}$ reverses its sign, which is observed in our experiment. In addition, the ratio between the coefficients $C_{\mathrm{MR}}$ and $C_{\mathrm{MS}}$ was found to be close to 2, which is consistent with the factor of 2 difference between equations (5) and (7). It can be seen from the deduction of equation (7) that this factor is 2 only if the power index of $E_F$ in $\Delta\sigma_{xx}$ is equal to the minus of that in the Drude conductivity, which is 2. Therefore, our results suggest that the chiral anomaly conductivity is inversely proportional to $E_F^m$ and $m$ is 2.

In summary, we have studied the low temperature thermoelectric effect of $Cd_3As_2$ micro-plates. When the field is perpendicular to the temperature gradient, the thermopower increases with the field, which is consistent with a simple one-band Drude transport. However, when the field is parallel to the temperature gradient, the thermopower displays an anomaly. It decreases with the magnetic field and even change its sign. In low fields, it is quadratically dependent of $B$, as the conductivity is, except that the coefficient is twice of that for the latter. Utilizing the Mott relation, we show that the observed anomalous thermopower of Weyl fermions can be well explained in terms of the chiral anomaly. Our results demonstrate the thermoelectric effect as an important technique for studying Weyl fermions and it may be used in study of other topological materials.

## Methods

**Sample growth.** $Cd_3As_2$ platelets were grown by a chemical vapour deposition method in a tube furnace. Precursor $Cd_3As_2$ powders were placed at the centre of the furnace, while silicon substrates coated with 5 nm gold film were placed downstream. Before the growth, the tube furnace was flushed several times with Argon gas. Then, an Argon flow of 20 s.c.c.m. was maintained as a carrier gas. The temperature was gradually raised to 650 °C and kept for 10 min. After the growth process, the furnace was cooled naturally. Structural characterization has been reported elsewhere, see the Supplementary Fig. 6 in ref. 14.

**Transport measurements.** The grown $Cd_3As_2$ platelets were transferred to a silicon substrate with an oxide layer of 285 nm. Devices were fabricated using the standard e-beam lithography, followed by deposition of 300 nm Au and lift-off. Electrical measurements were carried out in an Oxford cryostat using a lock-in method.

**Data availability.** The data that support the findings of this study are available from the corresponding author upon request.

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

## Acknowledgements

This work was supported by National Key Basic Research Program of China (No 2016YFA0300602, No. 2013CBA01603, No. 2012CB933404 and No. 2016YFA0300802) and NSFC (Project No. 11574005, No. 11222436 and No. 11234001).

## Author contributions

X.W. conceived the project. D.Y. and Z.L. provided suggestions and guidance for experiments. C.L. grew the samples and fabricated devices. Z.J. performed transport experiments with the help from X.L.; Z.J., J.S. and X.W. analysed the data. Z.J. and X.W. wrote the paper with help from all other co-authors.
