## [Peer Review File · Nature Communications]

Reviewers' comments:

Reviewer #1 (Remarks to the Author):

This manuscript reports measurements of the electrical resistivity and thermopower in the Dirac material Cd₃As₂. Under the application of a magnetic field (with different orientations with respect to an applied electric field and thermal gradient) transport signature of the chiral anomaly of Weyl fermions are obtained. (An applied magnetic field splits the 3D Dirac points into Weyl points in Cd₃As₂.)

Overall, I find the experimental results compelling and the presentation in the manuscript clear.

One point of possible confusion concerns the Fermi energy dependence of the chiral anomaly conductance. Eq.(4) shows $\frac{\sigma_{xx}(B)}{\sigma_{xx}(0)}=1+B^2/B_0^2$, where $B_0 \propto E_F^2$. Near the top of page 6, there are statements that the chiral anomaly conductance depends on the Fermi energy as $1/E_F^2$. The authors might want to move the $\Delta \sigma_{xx}$ result of Ref[9] near the bottom of page 4 to a more prominent location (displayed equation), since to get this from Eq.(4) one need $\sigma_{xx}(B=0) \propto E_F^2$.

Otherwise, I recommend publication.

Reviewer #2 (Remarks to the Author):

The authors report on studying electronic configurations of vapor-grown Cd₃As₂ by performing combined T-dependent resistivity, magnetoresistance (MR) and thermo-electric characterizations at various temperatures. In addition to the negative MR-effect, the sign reversal of thermopower and its parabolic dependence on the B-field (|| configuration) at the low-field limit are explained within the context of Weyl Fermion framework and related chiral anomaly. The work falls in a critically new area of solid-state physics/condensed matter, but in view of many deficiencies/ controversies it cannot be recommended for publication in Nature Comm.. Just a few examples:

1. Cadmium Arsenide can crystallize as both Cd₃As₂ and CdAs₂ crystals, whereas Cd₃As₂ features a very complex defective crystal structure and can undergo many polymorphic transitions. A tight control and detailed investigations of the crystal characteristics, including defects are thus paramount to successful analysis of the underlying electronic properties, excitation and charge-carrier dynamics. Likewise, the sample under study is incorrectly referred to nano-plate. Ref. # [14] is completely irrelevant as it deals with nanowires and not platelets.
2. Thermo-electric based measurement approach only adds another layer of complexity to the analysis as the heat diffusion of charge carriers has to be taken into account. The use of Mott equation is not well justified. The sign reversal of the thermopower can be easily traced back to the ambipolar charge carrier diffusion that was confirmed in many semiconductors including Cd₃As₂.
3. Several results, including resistivity vs. T data deviate significantly from those obtained in many other/ prior studies, in which the opposite trends were reported, see for instance supp. material part of APL 106, 231904 (2015). The study is based on the data obtained only from one sample and the results are unlikely to be broadly applicable.
4. The ms is not free of conflicting/incorrect statements. On p3. for instance, the authors

controversially state "The resistivity exhibits a metallic temperature dependence. As the temperature is raised, more carriers are thermally activated, which leads to reduction of resistivity."
5. It is surprising that authors failed to report any SdH oscillations in this work.

Reviewer #3 (Remarks to the Author):

The authors measured the thermoelectric power of the Dirac semimetal Cd₃As₂ plate. They have observed that the thermoelectric power increases with increasing magnetic fields and saturates at high fields when the magnetic field is perpendicular to the temperature gradient, but it is suppressed when magnetic field parallel to the temperature gradient and follows a quadratic dependence of magnetic field at small fields. The authors claim that the observations are related to the chiral anomaly in the Cd₃As₂. Although the results are interesting, the data presented are not sufficient and the analysis are not convincing. Therefore, I cannot recommend its publication in Nature Communication. Followings are some detailed comments.

1. The authors have claimed that at low temperatures holes are dominant carriers, while above 57 K thermally activated electrons become dominant. In addition to temperature dependent S_{xx} presented, the temperature dependent ordinary Hall effect measurements should be presented in the manuscript to verify this claim. The analysis of Hall data will allow the authors to obtain the temperature dependent carrier concentration and mobility for holes and electrons, respectively.
2. The authors have adopted single band Boltzmann-Drude transport model, i.e. $\sigma_{xx} = npe\mu/(1 + \mu^2B^2)$ and $\sigma_{xy} = npe\mu^2B/(1 + \mu^2B^2)$, to fit S_{xx} and S_{xy} in the perpendicular fields using Equation (2) and (3). The fitted magnetic field range for S_{xx} (> 5 T) and S_{xy} (< 1 T) is very different (Fig. 2b & 2c), why? In fact, the valid field range for Boltzmann-Drude transport model should be verified by magneto electrical transport measurements. In addition, the authors assume τ is independent of energy, but after breaking the time reversal symmetry by the external magnetic field the Fermi surface of Cd₃As₂ is rearranged and τ will change accordingly. The authors should discuss qualitatively the influence of τ on their fittings.
3. Only three calculated mobility μ (below 50 K) are shown in Fig. 2d because the two-band model for the thermoelectric simulation is too complex as the authors have claimed. In my view, the number of data points is too small to verify the good agreement between the experiments and theory.
4. The authors have claimed that at high temperatures electrons are dominating. I wonder if the authors can use the single band theory to analyze the results at high temperatures for example above 150 K.
5. In Fig. 3a and 3b, a dip around zero magnetic field is observed in both MR and S_{xx} curves with $B//E$, even at high temperature of 150 K, what is the physical origin of this dip?
6. The authors have only analyzed low temperature S_{xx} curves in Fig. 3b without paying any attention to high temperature S_{xx} curves. How do the authors explain S_{xx} dependence on magnetic field at higher temperatures?
7. It has been shown theoretically that different relative orientation of the thermal gradient, electric and magnetic field would induce different properties (Phys. Rev. B 90, 165115 (2014)). Would it be possible for the authors to change the orientation between the thermal gradients and the electric fields from parallel to perpendicular and measure the corresponding thermoelectric behaviors with $B \perp E$ and $B//E$?

Response to Reviewers' comments:

Reviewer #1:

This manuscript reports measurements of the electrical resistivity and thermopower in the Dirac material Cd₃As₂. Under the application of a magnetic field (with different orientations with respect to an applied electric field and thermal gradient) transport signature of the chiral anomaly of Weyl fermions are obtained. (An applied magnetic field splits the 3D Dirac points into Weyl points in Cd₃As₂.)

Overall, I find the experimental results compelling and the presentation in the manuscript clear.

One point of possible confusion concerns the Fermi energy dependence of the chiral anomaly conductance. Eq.(4) shows $\frac{\sigma_{xx}(B)}{\sigma_{xx}(0)}=1+B^2/B_0^2$, where $B_0 \propto E_F^2$. Near the top of page 6, there are statements that the chiral anomaly conductance depends on the Fermi energy as $1/E_F^2$. The authors might want to move the $\Delta \sigma_{xx}$ result of Ref[9] near the bottom of page 4 to a more prominent location (displayed equation), since to get this from Eq.(4) one need $\sigma_{xx}(B=0) \propto E_F^2$.

We thank the referee for his/her careful reading and the constructive advice, which we have adopted in the revision:

We display the expression of $\Delta \sigma_{xx}$ as an equation. We also use the notion σ_{xx}^0 for the Drude conductivity and $\Delta \sigma_{xx}^c$ for the conductivity correction due to the chiral anomaly and express the total conductivity as $\sigma_{xx}=\sigma_{xx}^0+\Delta \sigma_{xx}^c$.

Otherwise, I recommend publication.

Reviewer #2:

The authors report on studying electronic configurations of vapor-grown Cd₃As₂ by performing combined T-dependent resistivity, magnetoresistance (MR) and thermo-electric characterizations at various temperatures. In addition to the negative MR-effect, the sign reversal of thermopower and its parabolic dependence on the B-field (|| configuration) at the low-field limit are explained within the context of Weyl Fermion framework and related chiral anomaly. The work falls in a critically new area of solid-state physics/condensed matter, but in view of many deficiencies/ controversies it cannot be recommended for publication in Nature Comm.. Just a few examples:

1. Cadmium Arsenide can crystallize as both Cd₃As₂ and CdAs₂ crystals, whereas Cd₃As₂ features a very complex defective crystal structure and can undergo many polymorphic transitions. A tight control and detailed investigations of the crystal characteristics, including defects are thus paramount to successful analysis of the underlying electronic properties, excitation and charge-carrier dynamics. Likewise, the sample under study is incorrectly referred to nano-plate. Ref. # [14] is completely irrelevant as it deals with nanowires and not platelets.

We thank the referee for the careful reading and the comments. We realize that the description of our samples was too brief. We have in fact carried out extensive structural characterization on these samples, which confirms that they are Cd₃As₂. These characterizations have already been reported in our recently published paper, Ref. 14. In the paper, although nanowires were what were focused on, similar set of measurements were also carried out on micro-plates, for instance, Supplementary figure 6 in Ref. 14. These platelets are Cd₃As₂ and their electrical transport properties are very similar to nanowires. In fact, both were grown in the same batch. Therefore, we have modified the sentence in paragraph 3 on page, so the reference to these structural characterizations are clearly specified:

Structural characterization has been reported in elsewhere, see the Supplementary Figure 6 in Ref. 14.

We have changed nano-plate to micro-plate.

2. Thermo-electric based measurement approach only adds another layer of complexity to the analysis as the heat diffusion of charge carriers has to be taken into account. The use of Mott equation is not well justified. The sign reversal of the thermopower can be easily traced back to the ambipolar charge carrier diffusion that was confirmed in many semiconductors including Cd₃As₂.

We agree that thermoelectric effects are complicated. However, this is exactly what have surprised us when a simple model based on the chiral anomaly and the Mott relation had worked. It has made us believe that the model have captured the underling physics.

The Mott relation is widely used, even for 2D Dirac electrons in graphene (Ref. 36, 37, 38, 39) and 3D Dirac electrons in Pb_{1-x}Sn_xSe(Ref. 40). Nevertheless, its applicability to Weyl fermions in the condition of the chiral anomaly is not *a priori* knowledge. Our experiment indicates that the Mott relation works in this case. This is one of the conclusions of the current work.

We agree that the ambipolar charge carrier diffusion can give rise to a sign reversal, particularly when the contributions from two types of carriers are comparable. However, at low temperatures when thermal activation is suppressed, only one type of carriers is dominant. This has been discussed in the manuscript. In the revision, we have added in the supplementary material the Hall measurement at different temperatures, which confirms this, too. Furthermore, with increasing temperature, more electrons are excited and ambipolar transport is evident, as the referee suggested. But, its manifestation is an *increase* of the thermopower at low fields. This is because electrons has a negative magneto-thermopower due to the chiral anomaly, just like holes. The evolution of the ambipolar transport can be clearly seen in Fig. 3b. We have added a paragraph to discuss the effect.

We notice a dip in S_{xx} around zero field, see Fig.3b. It is absent at 10 K, while it emerges at 25 K and grows with temperature. The evolution is consistent with the contribution from thermally activated electrons. It is reasonable to believe that electrons also display the chiral anomaly induced negative magneto-thermopower. Due to the low density and high mobility, their contribution will be prominent at low fields, forming a dip around zero field. As the temperature increases, electrons

gradually gain its weight and eventually dominate S_{xx} . At 150 K, S_{xx} recovers a B^2 dependence, in agreement with Eq. 7.

Moreover, the sign reversal is observed in a parallel B, but not in a perpendicular B. To emphasize this point, we have modified the first part of the second paragraph on page 5, so it now reads:

At low temperatures ($T= 10, 25$ and 40 K) where one hole band dominates the transport, the thermopower decreases with the magnetic field. S_{xx} even drops to zero and change its sign. This anomalous field dependence is in sharp contrast to that in a perpendicular field, which we have shown can be explained by a dominant single band. Thus, the competition between two types of carriers is excluded as the origin of the anomalous field dependence.

3. Several results, including resistivity vs. T data deviate significantly from those obtained in many other/ prior studies, in which the opposite trends were reported, see for instance supp. material part of APL 106, 231904 (2015). The study is based on the data obtained only from one sample and the results are unlikely to be broadly applicable.

Depending on the carrier density, both a metallic and semiconducting behavior have been reported for Cd₃As₂. Bulk Cd₃As₂ materials often have a high electron density on the order of $10^{18}\sim 10^{19}/\text{cm}^3$, hence display a metallic behavior, see Ref. 44 and the one given by the referee. On the other hand, nanowires and micro-plates are hole-doped with a much lower density, on the order of $10^{17}/\text{cm}^3$, and display a semiconducting RT behavior, as reported by us (Ref. 14) and others (Ref. 18). A semiconducting behavior has also been observed in bulk Cd₃As₂ of low electron density, Ref. 45. Such a non-monotonic T-dependence is typical for semimetals of low carrier density, for instance, NaBi(Ref. 13). Therefore, our samples are typical. In the revision, we have also added data for another sample we had measured. Similar thermoelectric behaviors are observed.(Supplementary Figure S5 and S6 and the corresponding discussions)

4. The ms is not free of conflicting/incorrect statements. On p3. for instance, the authors controversially state "The resistivity exhibits a metallic temperature dependence. As the temperature is raised, more carriers are thermally activated, which leads to reduction of resistivity."

There is no confliction in the statement, but a non-monotonic RT behaviors. Before the statement, the context is "At low temperature, the carrier concentration is relatively constant. The resistivity exhibits a metallic...". The metallic dependence is in a low temperature regime, while thermal activation is in a high temperature regime. To avoid the confusion, we have modified that sentence:

At low temperatures, the carrier concentration is relatively constant and the resistivity exhibits a metallic temperature dependence.

5. It is surprising that authors failed to report any SdH oscillations in this work.

Absence of SdH has been commonly seen in nanowires and micro-plates of Cd₃As₂ and discussed (Ref. 14 and Ref. 18). As we have explained in reply to comment #3, nanowires and micro-plates are hole-doped and the carrier density is much lower than 3D bulk materials. Two effects possibly

contribute to the absence of SdH. First, the hole mobility of Cd3As2 is known to be orders of magnitude lower than that of electrons(Ref. 44). Second, the system quickly goes into the quantum limit when the carrier density is low. We have added a sentence in paragraph 3 on page 3, which refers readers to these previous work.

Shubnikov-de Haas oscillations are not discernible, likely due to the low mobility and low density of the hole band[14,18].

Reviewer #3:

The authors measured the thermoelectric power of the Dirac semimetal Cd3As2 plate. They have observed that the thermoelectric power increases with increasing magnetic fields and saturates at high fields when the magnetic field is perpendicular to the temperature gradient, but it is suppressed when magnetic field parallel to the temperature gradient and follows a quadratic dependence of magnetic field at small fields. The authors claim that the observations are related to the chiral anomaly in the Cd3As2. Although the results are interesting, the data presented are not sufficient and the analysis are not convincing. Therefore, I cannot recommend its publication in Nature Communication. Followings are some detailed comments.

1. The authors have claimed that at low temperatures holes are dominant carriers, while above 57 K thermally activated electrons become dominant. In addition to temperature dependent S_{xx} presented, the temperature dependent ordinary Hall effect measurements should be presented in the manuscript to verify this claim. The analysis of Hall data will allow the authors to obtain the temperature dependent carrier concentration and mobility for holes and electrons, respectively.

We thank the referee for taking time to read our manuscript and giving advices. We agree that the claim would be more solid if the Hall is shown and analyzed. So, we have added the Hall data of the sample into the supplementary materials of the revision. The results support our claim. We have added a sentence at the end of paragraph on page.

The switching of the dominant carrier is also evident in our Hall measurement(Supplementary Figure S2).

The evolution of the thermopower with temperature corroborates with the Hall measurement. We have added a paragraph to explain it. See the reply to comment #5.

2. The authors have adopted single band Boltzmann-Drude transport model, i.e. $\sigma_{xx} = npe\mu/(1 + \mu^2B^2)$ and $\sigma_{xy} = npe\mu^2B/(1 + \mu^2B^2)$, to fit S_{xx} and S_{xy} in the perpendicular fields using Equation (2) and (3). The fitted magnetic field range for S_{xx} (> 5 T) and S_{xy} (< 1 T) is very different (Fig. 2b & 2c), why? In fact, the valid field range for Boltzmann-Drude transport model should be verified by magneto electrical transport measurements. In addition, the authors assume τ is independent of energy, but after breaking the time reversal symmetry by the external magnetic field the Fermi surface of Cd3As2 is rearranged and τ will change accordingly. The authors should discuss qualitatively the influence of τ on their fittings.

We admit that using different fitted ranges for S_{xx} and S_{xy} is inappropriate. In the revision, we adopt the same range for both fits, $B < 1\text{T}$. The fitting is limited to a low field range to stay in the classical field regime, as was done in Ref. 40. For simplicity, we have considered a single band Boltzmann-Drude transport, without including the field dependence of τ . However, Cd_3As_2 often exhibits a linear MR, indicating a B dependent τ , as suggested by the referee. So, we have estimated its influence on fittings, by explicitly introducing a linear B term in τ and deriving the expression for S_{xx} and S_{xy} . Note that the magnetoresistance of our sample is 30-50% at 1 T and low temperatures. It turns out that the fitting result only slightly changed. We have presented the derivation and the fittings to the new equation in the supplementary materials. For completeness, a discussion on the influence of the energy dependence of τ has been provided, too. We have added a few lines at the end of the first paragraph on page 4 to refer to the corresponding discussion.

It is worth mentioning that taking into consideration a magnetic dependent τ , as suggested by the linear magnetoresistance of Cd_3As_2 in Fig. 2a, only slightly changes the fittings (see detailed discussion in the supplementary materials).

3. Only three calculated mobility μ (below 50 K) are shown in Fig. 2d because the two-band model for the thermoelectric simulation is too complex as the authors have claimed. In my view, the number of data points is too small to verify the good agreement between the experiments and theory.

It is true that more points will make the argument more convincing. We wish we could provide more points. But, we'd like to point out that each of these six points are the results of fitting of six curves and the fits are all very good. Moreover, a good agreement has already been found for other Dirac systems, including 2D (graphene, Ref. 36, 37) and 3D ($\text{Pb}_{1-x}\text{Sn}_x\text{Se}$, Ref. 40). Our fitting results confirm these previous work. At last, the main result of the current work, the chiral anomaly induced thermopower in a parallel field, is not undermined.

4. The authors have claimed that at high temperatures electrons are dominating. I wonder if the authors can use the single band theory to analyze the results at high temperatures for example above 150 K.

At 150 K, a negative magneto-thermopower is also observed and proportional to B^2 , as seen in the supplementary Fig. S4. However, the B^2 coefficient 0.0077 is significantly smaller than that for σ_{xx} , 0.0386. At this time, we are not sure about the reason. Possible candidates are the thermal smearing effect on the chiral anomaly and the change of the scattering mechanism, hence the energy dependence of τ , for instance, from impurity scattering to phonon scattering. We have discussed the high temperature thermopower in an added paragraph on page 6.

At 150 K, S_{xx} displays a B^2 dependence, in agreement with Eq. 7. However, the coefficient ratio - C_{MS}/C_{MR} is 0.2, significantly smaller than 2. The reason is not clear, but it might be related to thermal smearing of the chiral anomaly or electron-electron scattering or electron phonon scattering, see the supplementary materials.

5. In Fig. 3a and 3b, a dip around zero magnetic field is observed in both MR and S_{xx} curves with $B//E$, even at high temperature of 150 K, what is the physical origin of this dip?

We are grateful for the referee's comment. After we had taken a close look on the feature, we realized that the dip is caused by the contribution from electrons. Its evolution with temperature agrees well with thermal activation, which further strengthens our arguments. We explain it as follows. The dip in MR is not significant and only appear at 150 K. The data are noisier than other temperatures. So, we believe that it is most likely an artifact. Let's focus on the dip in S_{xx} , which shows a systematic change with temperature. The dip is absent at 10 K. And then, it grows with increasing temperature. According to the Hall measurement, we have more and more thermally excited electrons as the temperature increases, hence their contribution to S_{xx} . It is reasonable to believe that the thermopower of electrons also has a chiral anomaly induced negative field dependence, just as holes. Since electrons are of low density and high mobility, their contribution will be prominent at low fields. So, the contribution of electrons will give rise to a dip around zero field, as we observed. At high temperatures, electrons dominate the thermopower. The dip grows and becomes a clear manifest of the chiral anomaly induced negative magneto-thermopower with an overall negative sign. On the top of page 6, we have added a paragraph to explain the feature:

We notice a dip in S_{xx} around zero field, see Fig.3b. It is absent at 10 K, while it emerges at 25 K and grows with temperature. The evolution is consistent with the contribution from thermally activated electrons. It is reasonable to believe that electrons also display the chiral anomaly induced negative magneto-thermopower. Due to the low density and high mobility, their contribution will be prominent at low fields, forming a dip around zero field. As the temperature increases, electrons gradually gain its weight and eventually dominate S_{xx} . At 150 K, S_{xx} recovers a B^2 dependence, in agreement with Eq. 7.

6. The authors have only analyzed low temperature S_{xx} curves in Fig. 3b without paying any attention to high temperature S_{xx} curves. How do the authors explain S_{xx} dependence on magnetic field at higher temperatures?

See the reply to comment #4 and #5.

7. It has been shown theoretically that different relative orientation of the thermal gradient, electric and magnetic field would induce different properties (Phys. Rev. B 90, 165115 (2014)). Would it be possible for the authors to change the orientation between the thermal gradients and the electric fields from parallel to perpendicular and measure the corresponding thermoelectric behaviors with $B \perp E$ and $B//E$?

It is indeed a very interesting, yet challenging if not impossible, experiment. In a thermoelectric measurement setup, a temperature gradient ∇T is applied and the response, E , is measured. So, ∇T is always parallel to E , hence parallel to B . If one wants to have $B \perp E$, he needs to short the longitudinal voltage and measure the current response, and apply a transverse electrical field. Considering the inevitable contact resistances and many other issues, these tasks are highly nontrivial, especially when the sample resistance is not very high. We have not seen such an

experiment. This is an issue often encountered when one tries to compare experiments with theories. Theories often calculate the current/conductivity, while experiments measure the voltage/resistivity.

Reviewers' comments:

Reviewer #1 (Remarks to the Author):

The other two referees have raised a number of important technical issues, particularly the third referee. On the whole, I think the authors have done a good job addressing these issues. The main weaknesses of the work at this point appear to be the relatively small number of samples that have produced "good data" and the theoretical analysis of the data, which is restricted to certain temperature regimes (mainly the low-temperature regime). In spite of these weakness, I think the work is a substantial improvement over the previous version and presents enough "meat" to be of interest to the broader community--both for experimentalists and theorists.

If the Weyl or Dirac materials are to find their way into some application, it seems most likely that it will be a bulk transport-related property that will be crucial (as opposed to ARPES measured Fermi arcs on the surface). The current work advances our understanding of the temperature and magnetic field dependence of electrical and thermal transport in Dirac materials. Particularly, with the improvements to the manuscript, and the additional material added to the supplementary information in response to the referees, I recommend publication.

Reviewer #2 (Remarks to the Author):

The authors have put an effort in revising their work, but as the changes were mostly formal and as the manuscript is still not free of controversy, it cannot be recommended for publication in Nature Communications.

The ms remains very vague in what exactly the main aim of the study is: is it studying a thermo-electric response of Weyl semi-metals or b) identification of the latter based on thermo-electric measurements (less important) ? If it is a), the authors have to prove beyond a reasonable doubt that they have Weyl semi-metal in the first place by performing additional, extensive characterizations/verifications. Negative MR is not a clear-cut indicator of Weyl semi-metals as negative MR can be induced by other effects in highly doped semiconductors and metals, including weak-localizations. (arXiv:cond-mat/9606108).

Since Cd₃As₂ solidifies as a crystal, the response is expected to be anisotropic and the starting point is in fact careful characterization of the samples. The claim that nanowires and platelets that are to grow by different routes/under different (local) conditions will share the same properties is highly unconvincing.

Fig S2, c shows electron density becoming negative at $T > 150\text{K}$, which is not technically feasible.

Likewise, some of the replies to the concerns raised by the referees came short of expectations/are

controversial and confusing. For instance, Ref.# 3 in the comment 2 raises concern on τ remaining energy dependent in the presence of a magnetic field, whereas the authors agreeing on this in general, in the conclusion claim the opposite, i.e. they "see only slightly changes in the fitting results". There is a difference between " τ being energy and B dependent"--the authors seem to be confused about this aspect.

The authors state that at low T "the carrier concentration is relatively constant", whereas the carrier densities in fact show linear-like dependencies on T, even at low T.

The authors failed to give a clear justification behind using Mott-equation in their analysis.

The graphs/ data are plotted without error-bars.

The work falls more into Phys. Rev. B (PRB) and less into Nature Communications territory, and the authors are recommended to submit their ms to PRB instead.

Reviewer #3 (Remarks to the Author):

The authors have addressed my comments satisfactorily. The manuscript can be published.

Response to Reviewers' comments:

Reviewer #2:

The authors have put an effort in revising their work, but as the changes were mostly formal and as the manuscript is still not free of controversy, it cannot be recommended for publication in Nature Communications.

The ms remains very vague in what exactly the main aim of the study is: is it studying a thermo-electric response of Weyl semi-metals or b) identification of the latter based on thermo-electric measurements (less important) ? If it is a), the authors have to prove beyond a reasonable doubt that they have Weyl semi-metal in the first place by performing additional, extensive characterizations/verifications. Negative MR is not a clear-cut indicator of Weyl semi-metals as negative MR can be induced by other effects in highly doped semiconductors and metals, including weak-localizations. (arXiv:cond-mat/9606108).

The chiral anomaly of Weyl fermions has already been demonstrated in Cd_3As_2 through a negative MR, Nature Comms. 6, 10137(ref. 14) and Nature Comms. 7, 10301(ref. 18). Our MR is consistent with those earlier studies. Therefore, there is no need to repeat those studies. As for the weak-localization, there are also detailed discussions in these earlier studies, which have carefully ruled out the weak-localization as the origin of the negative MR.

In the first paragraph on page 5, we have add a sentence to emphasize this point. It now reads:

Thus, the Dirac semimetal turns into a Weyl semimetal. This has been experimentally demonstrated in Cd₃As₂[14,18].

At the end of this paragraph, it has been modified as

All these behaviours have been discussed in our previous studies[14] and consistent with others[18].

As explained in the introduction, only two pieces of experimental evidence for Weyl fermions have been established so far, i.e. Fermi arcs (by ARPES) and a negative MR, while other properties remain to be explored. Many studies have used a negative MR as evidence for Weyl fermions, ref.12-21. We want to point out that experimental investigation of Weyl fermions has just begun. Negative MR is probably not a smoking gun for Weyl fermions. Hopefully, the current work will provide another means to test existence of Weyl fermions in new materials.

Since Cd₃As₂ solidifies as a crystal, the response is expected to be anisotropic and the starting point is in fact careful characterization of the samples. The claim that nanowires and platelets that are to grow by different routes/under different (local) conditions will share the same properties is highly unconvincing.

Our experiments have shown that nanowires and platelets share the same properties. They have the same crystal structure (see supplementary fig.6 in ref. 14). They have similar RT behavior (Ref. 14, 18 and this work). They have similar positive magnetoresistance for B_{\perp} and similar negative magnetoresistance for B_{\parallel} (Ref. 14 and 18).

Fig S2, c shows electron density becoming negative at $T > 150\text{K}$, which is not technically feasible.

It is a matter of definition. The negative sign of the electron density was to denote the negative charge of electrons. Since the legend has already labeled the curve for electrons as n_e , we remove the negative sign in the revision to avoid possible confusion.

Likewise, some of the replies to the concerns raised by the referees came short of expectations/are controversial and confusing. For instance, Ref.# 3 in the comment 2 raises concern on τ remaining energy dependent in the presence of a magnetic field, whereas the authors agreeing on this in general, in the conclusion claim the opposite, i.e. they "see only slightly changes in the fitting results". There is a difference between " τ being energy and B dependent"--the authors seem to be confused about this aspect.

We have explained in the previous reply that the magnetoresistance of our sample is 30-50% at 1 T. The change of the resistance, hence τ , is simply not large enough to change the fitting substantially. Supplementary Figure 3 and Note 4 are devoted to this.

The effect of both the energy dependence and B dependence of τ have been discussed in the Supplementary Note 3 "Effect of the energy dependence of τ on thermoelectric effects" and Note 4 "Effect of the field dependence of τ on thermoelectric effects".

The authors state that at low T "the carrier concentration is relatively constant", whereas the carrier densities in fact show linear-like dependencies on T, even at low T.

The positive temperature coefficient of resistivity at low temperatures indicates that the increase of the carrier density due to thermal activation is not strong enough to qualitatively change the metallic behavior. In the revision, we change "the carrier concentration is relatively constant" into "the increase of the carrier density is not significant". This is a common feature observed in semimetals of low carrier density and well understood (ref. 13, 14, 18 and 45).

The authors failed to give a clear justification behind using Mott-equation in their analysis.

As the Mott relation has been widely used, even in 2D and 3D Dirac systems(Ref. 36-40), it is fairly reasonable to attempt to apply it to Weyl semimetals. It turned out that it worked. It well explains our thermoelectric experiments. We therefore drew a conclusion that the Mott relation is applicable in Weyl semimetals.

The graphs/ data are plotted without error-bars.

We have added error-bars to Fig. 2d.

The work falls more into Phys. Rev. B (PRB) and less into Nature Communications territory, and the authors are recommended to submit their ms to PRB instead.

Other changes we have made:

Three references(17,20,21) have been updated to reflect the current state of publication.

The abstract has been slightly modified to comply with the 150-word limit.

The main text has been divided into sections.

Sections in the Supplementary Material have been re-ordered according to the order in which they are referred to in the main text.

And many other small changes have been made to comply with the format requirements of Nature Communications according to the manuscript checklist.

Reviewers' comments:

Reviewer #3 (Remarks to the Author):

[Note from the editor: the referee provided confidential comments to the editor, approving the authors' responses.]